# Gradient Flow Convergence Guarantee for General Neural Network Architectures

## Abstract

A key challenge in modern deep learning theory is to explain the remarkable success of gradient-based optimization methods when training large-scale, complex deep neural networks. Though linear convergence of such methods has been proved for a handful of specific architectures, a united theory still evades researchers. This article presents a unified proof for linear convergence of continuous gradient descent, also called gradient flow, while training any neural network with piecewise non-zero polynomial activations or ReLU, sigmoid activations. Our primary contribution is a single, general theorem that not only covers architectures for which this result was previously unknown but also consolidates existing results under weaker assumptions. While our focus is theoretical and our results are only exact in the infinitesimal step size limit, we nevertheless find excellent empirical agreement between the predictions of our result and those of the practical step-size gradient descent method.

## 1 Introduction

Behind training any successful deep learning model is the hope of achieving a low loss. In this article, we turn that hope into certainty, proving that gradient flow based training does not just succeed, but it converges exponentially fast, thus proving that training procedures need not be a gamble.

Understanding why gradient-based optimization methods are so effective at training large, complex deep neural networks is a central question in modern machine learning theory. Despite the non-convex nature of the loss landscape, these methods can find solutions with low training error. This empirical success is well-known and is a reason for the success of neural networks in various real-life tasks. However, finding theoretical guarantees that training via gradient methods converges, especially at exponential rates[1] and across a broad class of architectures, remains an active area of research. Such guarantees are not only of theoretical interest but also provide valuable insights into why neural networks generalize well and how to design better models.

As we discuss below, a number of works have studied linear convergence of gradient methods for specific network architectures, initialization schemes or under strong over-parameterization assumptions. However, a unifying framework that applies across diverse architectures and does not rely on restrictive assumptions has been missing.

In this article, we close that gap by proving that training neural networks using continuous gradient descent will almost always lead to an exponential decay of the training loss, ie. we prove that at time $t$ of applying continuous gradient descent, loss is $\mathcal{O}(e^{-t})$. An informal version of our result is as follows.

**Theorem 1.1** (Informal theorem). *Let $P$ be the number of trainable parameters of a neural network, $n$ be the training dataset size and $M$ be the output dimension. If the network is over-parametrized, ie. $P \geq nM$ and the initialization, input data distributions are absolutely continuous, then linear convergence of loss holds with probability 1.*

---

[1]Linear convergence of loss and exponential decay of loss mean the same thing. This is common terminology used in optimization - linear convergence of loss means that the loss $L$ decreases linearly, ie. $L_t \leq \alpha L_{t-1}$ for some $\alpha \geq 0$. Iterating backwards, we get $L_t \leq \alpha^t L_0$.

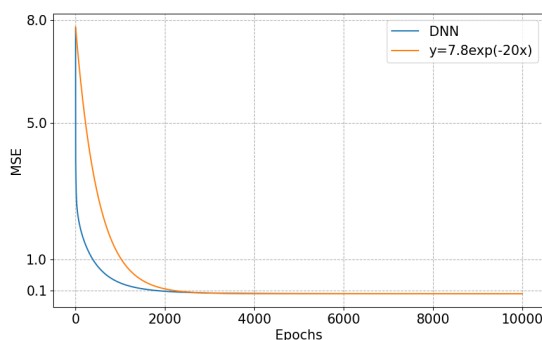

Figure 1: The error curve for a deep neural network trained on random data is bounded by an exponential decay. Thus, the error here is $\mathcal{O}(e^{-t})$.

In other words, as long as the dataset and initialization are not pathological, and the network has sufficient capacity to represent the task, GF (gradient flow) leads inevitably to exponential loss decay. A larger output dimension or larger sample size means the learning is potentially much more complex, and thus needs more parameters to be learnt. This conclusion is aligned with the well known empirical claim that over-paramterized networks perform better.

To do so, we track the training dynamics using the NTK, as done by previous works (Du et al., 2019b;a) and use properties of ODEs (Chatterjee, 2022).

Our novel contribution lies in realizing that the set of initializations for which the NTK has a zero eigenvalue has zero probability with respect to initialization and input data distributions, and that this property is preserved throughout training. Once positive-definiteness of the NTK has been established, routine calculations lead us to an exponential decay of the loss.

Compared to existing work, our proof is both more general and simpler. The argument avoids lengthy technical calculations, requiring only basic probability and ODE theory. Yet it yields a result that appears to be absent from the literature.

One subtelty arises from activation functions. Our proof directly applies to piecewise nonzero polynomial activations, which excludes the widely used ReLU and sigmoid functions. To solve this problem, we use a piecewise non-zero polynomial activation that uniformly lower bounds ReLU/sigmoid and use a limiting argument to show that if this activation converges uniformly to ReLU/sigmoid, then exponential decay holds for ReLU/sigmoid as well.

**Applicability of our work**    Theorem 3.1 holds for a general function structure composed of polynomial layers and a piecewise nonzero polynomial activation function, which covers many popular architectures like DNNs, ResNets (He et al., 2016), CNNs (Krizhevsky et al., 2012), ConvResNets (Du et al., 2019a), GCNs (Kipf & Welling, 2017), ChebConvNets (Defferrard et al., 2016), Graph-ConvNets (Morris et al., 2019), U-Nets (Ronneberger et al., 2015), MobileNets (Howard et al., 2017) and SiameseNets (Taigman et al., 2014) with activations like leaky ReLU (Maas et al., 2013) or parametric ReLU (He et al., 2015), just to name a few. Note that corollaries 3.1.1, 3.1.2 extends our conclusion to ReLU, sigmoid activations as well. Moreover, most common initializations and input data are sampled from an absolutely continuous distribution. Thus, our work holds for most of the practically relevant architectures, input data and initializations.

### 1.1    COMPARISON WITH PREVIOUS WORKS

The seminal work by Jacot et al. (2018) proved that the training of DNNs with a smooth activation function in the infinite width limit for a finite time with gradient descent can be characterized by a kernel. Our proof technique relies on the same kernel matrix used by them.

Linear convergence of training error has been known under simple settings. Some works that use a trajectory-based analysis of the algorithm dynamics are compared in table 1 along important as-

Table 1: Assumptions and conclusions of previous works. 'w' stands for architectures without biases, 's' for shallow, 'L' for layers, $L$ for width of the network, $n$ for training dataset size, $T$ for training time, $E, V$ for the number of edges and vertices of the graph used for GNN, D for DNN, C for CNN, R for ResNet, CR for CNN-ResNet, G for GNN, pl for polylog, p for poly and e for error. We write etc. whenever there are too many specific conditions (or cases) that need to be satisfied (or can be applied to), and instead just mention the most restrictive (or the most common) one. Specific refers to a condition that is fairly specific and cannot be condensed into one word.

| Paper | GD or GF | Architectures | Activation | Rate |
|---|---|---|---|---|
| Allen-Zhu et al. (2019a) | GD | D (3L,w) | ReLU | $1/\sqrt{T}$ |
| Allen-Zhu et al. (2019b) | GD | D,R,C (w) | ReLU | $e^{-T}$ |
| Du et al. (2019a) | GD | D,R,CR (w) | Analytic, etc. | $e^{-T}$ |
| Nguyen & Mondelli (2020) | GD | D (w) | Smooth, sub-lin. | $e^{-T}$ |
| Zou et al. (2020) | GD | D (w) | ReLU | $1/T$ |
| Awasthi et al. (2021) | GD | G (s) | ReLU | $e^{-T}$ |
| Chen et al. (2021) | GD | D (w) | ReLU | $1/T$ |
| Liu et al. (2022) | GD | D (w) | Twice diffn. | $e^{-T}$ |
| Gopalani & Mukherjee (2025) | GD | D (s,w) | Softplus | $e^{-T}$ |
| Du et al. (2019b) | GF&GD | D (s,w) | ReLU | $e^{-T}$ |
| Chatterjee (2022) | GF&GD | D | $C^2$ | $e^{-T}$ |
| Boursier et al. (2022) | GF | D (s,w) | ReLU | $e^{-T}$ |
| This paper | GF | Any | ReLU, $\sigma$, etc. | $e^{-T}$ |

| | Hidden size | Initialization | Training Data | Out.dim. |
|---|---|---|---|---|
| Allen-Zhu et al. (2019a) | $\mathcal{O}(1/e)$ | Normal | Normalized | Multi |
| Allen-Zhu et al. (2019b) | $p(L,n)$ | Normal | Separable | Multi |
| Du et al. (2019a) | $\mathcal{O}(n^4)$ | Arbitrary | Normalized | Single |
| Nguyen & Mondelli (2020) | n | Normal, etc. | Almost any | Multi |
| Zou et al. (2020) | $\Omega(n^{14})$ | Normal | Separability | Single |
| Awasthi et al. (2021) | Arbitrary | Normal | $E = o(\sqrt{V})$ | Multi |
| Chen et al. (2021) | $pl(1/e)$ | Normal | $n=\mathcal{O}(1/e)p(L)$ | Single |
| Liu et al. (2022) | $\mathcal{O}(n)$ | Normal | Arbitrary | Multi |
| Gopalani & Mukherjee (2025) | Arbitrary | Arbitrary | Arbitrary | Single |
| Du et al. (2019b) | $\mathcal{O}(n^6)$ | Normal | Non-parallel | Single |
| Chatterjee (2022) | Arbitrary | Arbitrary | Linear ind. | Single |
| Boursier et al. (2022) | Specific | Nonzero | orthogonality | Multi |
| This paper | $P \geq nM$ | Almost any | Almost any | Multi |

sumptions and conclusions. It is to be noted that despite ease of analysis with GF, most of the previous works do not apply to GF. We have closed this gap with this work.

Chatterjee (2022) is one of the closest to our work. It proves that for DNNs with twice differentiable activations trained on a linearly independent input training dataset with either GF or GD with small step size, convergence is linear. However, their assumptions are yet again much more restrictive in comparison to ours.

The works by Min et al. (2023); Bah et al. (2022); Tarmoun et al. (2021); Xu et al. (2025) proved linear convergence of GF/GD, but for linear networks (ie. networks with linear activations, essentially making it linear regression). Hutzenthaler et al. (2023) provides a detailed analysis of convergence of both GF and GD, but for constant target functions.

Other approaches to finding convergence properties include landscape-based analysis (Arora et al., 2022; Jin et al., 2017), mean-field approximation based analysis (Chen et al., 2022; Ding et al., 2022), using optimal transport theory (Chizat & Bach, 2018; Khamis et al., 2024) and studying models in the richer albeit challenging 'feature learning regime' (Yang & Hu, 2020; Kunin et al., 2024).

**Why Gradient Flow ?** We focus on gradient flow to not have to keep track of discrete step sizes, which complicates the analysis and pulls attention away from the more important insights. More-

over, Elkabetz & Cohen (2021) proves that GF is close to GD with small step sizes for DNNs with homogeneous activations. It is to be noted that even though analysis using GF is usually less cumbersome than GD, the proof in this paper is not anywhere in the literature - despite the simplicity of its arguments. Previous works do talk about the convergence of GF, but they are again limited to either shallow networks (Du et al., 2019b; Boursier et al., 2022; Gopalani & Mukherjee, 2025; Tarmoun et al., 2021) or deep networks with smooth (Chatterjee, 2022) or linear activations (Min et al., 2023; Bah et al., 2022; Xu et al., 2025). As stated before, we nevertheless back up our claims empirically by training with GD.

**Our Novelty** The major novelty of our result is that it holds for a large variety of network architectures. Previous results have often been found for bias-less architectures (Du et al., 2019b; Allen-Zhu et al., 2019; Liu et al., 2022), one-dimensional (Gopalani & Mukherjee, 2025; Du et al., 2019a) or constant (Hutzenthaler et al., 2023) targets, very high over-parameterization (Du et al., 2019a; Allen-Zhu et al., 2019a; Chen et al., 2021), a specific initialization of weights (Liu et al., 2022; Zou et al., 2020; Nguyen & Mondelli, 2020) or strong assumptions on input data like orthogonality (Boursier et al., 2022) or linear independence (Chatterjee, 2022). Such convergence guarantees have also been found for other architectures like GNNs (Awasthi et al., 2021), CNNs (Allen-Zhu et al., 2019b) and ResNets (Du et al., 2019a). However, there has not been one work that combines all these results (and more) into a single, easy to follow theorem.

Note that though there have been guarantees for ReLU or sigmoid networks, there has been no work regarding leaky ReLU or parametric ReLU activated networks in the past. Our result holds for, as discussed before, a plethora of networks. We thus keep our experiments in section 4 limited to the more popular architectures.

To summarise, our **major contributions** are as follows:

- We prove linear convergence of gradient flow for a wide variety of architectures. Previous analyses have worked out linear convergence on a case-by-case basis (Awasthi et al., 2021; Bah et al., 2022). We prove a general theorem that works for any general function structure composed of polynomial layers and any piecewise non-zero polynomial activation, or ReLU/sigmoid activation.

- We require initialization to be from any absolutely continuous distribution. Previous analyses usually have only considered normal distributions (Du et al., 2019a; Nguyen & Mondelli, 2020).

- We require overparametrization of only at least $nM$. Previous works have required over-parameterization to be of much larger magnitudes (Du et al., 2019b; Chen et al., 2021).

Despite being a rather short and concise proof, our contributions sharpen, generalize, and in some cases correct limitations in prior work by covering a wide range of architectures with weaker assumptions on activation functions, initialization, training data and over-parameterization.

## 2 SETUP AND PRELIMINARIES

Consider a continuous function $f : \mathbb{R}^N \times \mathbb{R}^P \to \mathbb{R}^M$ defined as follows for some $N, P, M \in \mathbb{N}$:

$$f(X, \theta) := g_L(\sigma(\ldots g_2(\sigma(g_1(X, \theta)))\ldots)) \tag{1}$$

for all inputs $X \in \mathbb{R}^N$ and parametrizations $\theta \in \mathbb{R}^P$, where $g_i : \mathbb{R}^{N_{i-1}} \times \mathbb{R}^P \to \mathbb{R}^{N_i}$ are polynomials (in each component) for all $i \in [L]$ with $N_0 = N, N_L = M$ and $N_1, \ldots, N_{L-1} \in \mathbb{N}$ for some $L \in \mathbb{N}$. $\sigma : \mathbb{R} \to \mathbb{R}$ is a continuous, piecewise polynomial (with finitely many closed pieces) which is zero at finitely many values only. Note that $\sigma$ applied to a vector means that it is applied component-wise.

We shall prove our result for this function. We do so because most of the neural network architectures can be represented by this function. Examples will be given in corollary 3.2.1.

We intend to find that parametrization of $f$ that minimizes its error over a training dataset $(X_1, y_1), \ldots, (X_n, y_n) \in \mathbb{R}^N \times \mathbb{R}^M$, where $L : \mathbb{R}^P \to [0, \infty)$ is the error, defined by $L(\theta) := \frac{1}{2} \sum_{j=1}^{n} ||f(X_j, \theta) - y_j||^2$ for all $\theta \in \mathbb{R}^P$.

To do so, we utilize gradient flow. Start from some initial parameters $\theta_0$ and input data $X_1, \ldots, X_n$ sampled from some arbitrary absolutely continuous probability distributions (with respect to the Lebesgue measure). Then, for some $T > 0$, solve the following initial-value problem till $t = T$ to get the final parametrization $\theta(T)$.

$$\frac{\mathrm{d}\theta(t)}{\mathrm{d}t} = -\frac{\partial L(\theta(t))}{\partial \theta(t)} \tag{2}$$

$$\theta(0) = \theta_0$$

We shall often write $\nabla L(\theta(t))$ in place of $\frac{\partial L(\theta(t))}{\partial \theta(t)}$. Note that the $f$ need not be differentiable with respect to $\theta$ everywhere, but it is differentiable almost everywhere, due to the definition of $\sigma$. Note that this is will not be an issue, since the ODE theorems are applied piecewise and the training dynamics are not affected by almost everywhere differentiability.

The following lemma will prove that this ODE has a unique continuous and piecewise differentiable solution that can be extended till any $T > 0$.

**Lemma 2.1** (Existence and invertibility of GF solution curve). *The GF initial-value problem (2) has a unique, continuous and piecewise differentiable solution that can be extended till any $T > 0$, for almost all initializations and input data. Moreover, the solution flow is a diffeomorphism over the parameter space minus a measure zero set.*

*Proof.* Note that we assume $\nabla L(\theta(0)) \neq 0$ to exclude the trivial case of constant solution. Since GF cannot reach such a critical point, we need not worry about this case for the inverse solution. Moreover, we assume the initialization does not occur on a boundary, in order to apply Picard's theorem. Since boundaries and the set $\{\theta \in \mathbb{R}^P : \nabla L(\theta) = 0\}$ have zero measure (since $\nabla L(\theta)$ is a piecewise nonzero polynomial for almost all input datasets), the following arguments hold for almost all intializations amd input data.

Consider $-\nabla L(\theta)$ in its initialization piece. Then, it is a polynomial in $\theta$, and thus is Lipschitz continuous in any closed ball around the initialization $\theta_0$. By Picard's existence and uniqueness theorem (Hartman, 1964) the GF equation has a unique solution in a neighbourhood of $t = 0$. Since there is no finite time blowup of the solution (see theorem B.3), it can be extended beyond a neighbourhood of $t = 0$ (Hartman, 1964), either to all $t > 0$ or till it hits the boundary of its piece.

If this solution hits the boundary of the piece containing the initial condition $\theta(0)$, we can construct solution for the same ODE in the next piece, with initial condition being defined as the end point of the previous solution curve. Note that the curve will reach an intersection of more than two pieces for a zero measure set of initializations (see theorem B.1). Consequently, we do not need to worry about having to choose between two or more pieces for the solution curve to continue into.

Finally, these solutions can be joined together continuously (due to no finite time blowup, see theorem B.3) until we reach $t = T$. Define this piecewise differentiable, but continuous everywhere solution of the GF equation by $\Phi_t(\theta_0) := \theta(t)$ for all $t \geq 0$.

Now, note that the $\Phi_t$ is a piecewise diffeomorphism over the initial parameter space minus a measure zero set (ie. excluding critical points, boundary points and points with image/preimage curve that reaches an intersection of more than two pieces) since $-\nabla L(\theta(t))$ is piecewise continuously differentiable with respect to initial conditions (Hartman, 1964), all piecewise solutions are invertible as they are solutions of an autonomous system of ODEs, which all exist and can be joined uniquely by the above arguments, and the inverse solution $\Phi_t^{-1} = \Phi_{-t_0}$ is also piecewise differentiable (again by previous arguments). □

## 3 MAIN RESULT

**Theorem 3.1** (Linear convergence of GF). *Appling gradient flow to functions of form (1) with $P \geq nM$, the following holds almost surely with respect to the initialization and input distributions, for all $t \in [0, T]$.*

$$||y - F_t|| \leq e^{-\lambda_0 t} ||y - F_0|| \tag{3}$$

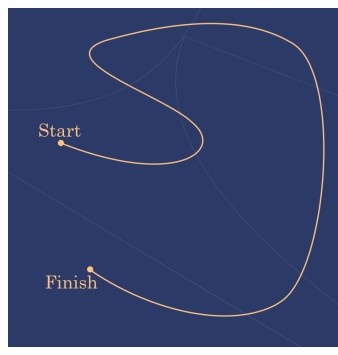 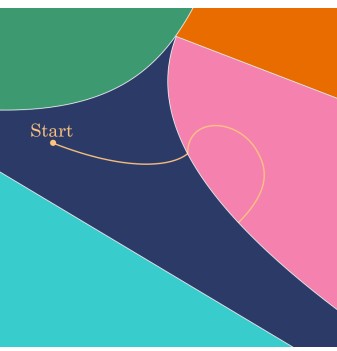 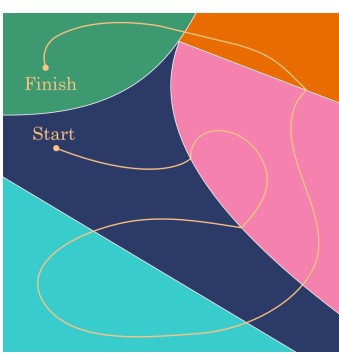

(a) Solution curve of the ODE in the initialization (blue) piece till finish ($t = T$). We cut off this solution at the point where it exits the initialization piece.

(b) We continue the solution by solving the ODE in the next (pink) piece and continue till we exit it.

(c) We keep on finding solutions for every piece and join them together to form the final solution. The curve will not hit an intersection of more than two pieces for almost all initializations.

Figure 2: Constructing the solution curve through the parameter space.

*which is equivalent to saying $L(\theta(t)) \leq e^{-2\lambda_0 t} L(\theta(0))$, where $\lambda_0 > 0$ and*

$$F_t := ((f(X_1, \theta(t)))_1, \ldots, (f(X_1, \theta(t)))_M, \ldots, (f(X_n, \theta(t)))_1, \ldots, (f(X_n, \theta(t)))_M)^T \in \mathbb{R}^{nM}$$

$$y := ((y_1)_1, \ldots, (y_1)_M, \ldots, (y_n)_1, \ldots, (y_n)_M)^T \in \mathbb{R}^{nM}$$

*Proof.* The first half of this proof is inspired by previous works (Du et al., 2019a;b), where we bound the error using the least eigenvalue of the 'neural tangent kernel'.

Fix some $t \geq 0$. Gradient of loss can be written as $\frac{\partial L(\theta(t))}{\partial \theta(t)} = \sum_{j=1}^n \frac{\partial f(X_j, \theta(t))}{\partial \theta(t)}(f(X_j, \theta(t)) - y_j)$ by chain rule. Using this fact, we calculate the dynamics of the predictions.

$$\frac{\mathrm{d}}{\mathrm{d}t} f(X_i, \theta(t)) = \left(\frac{\partial f(X_i, \theta(t))}{\partial \theta(t)}\right)^T \frac{\mathrm{d}\theta(t)}{\mathrm{d}t} = -\left(\frac{\partial f(X_i, \theta(t))}{\partial \theta(t)}\right)^T \frac{\partial L(\theta(t))}{\partial \theta(t)}$$

$$= -\left(\frac{\partial f(X_i, \theta(t))}{\partial \theta(t)}\right)^T \left(\sum_{j=1}^n \frac{\partial f(X_j, \theta(t))}{\partial \theta(t)}(f(X_j, \theta(t)) - y_j)\right)$$

$$= \sum_{j=1}^n \left(\left(\frac{\partial f(X_i, \theta(t))}{\partial \theta(t)}\right)^T \frac{\partial f(X_j, \theta(t))}{\partial \theta(t)}\right)(y_j - f(X_j, \theta(t))) \quad (4)$$

Concatenating output vectors, we get $\frac{\mathrm{d}F_t}{\mathrm{d}t} = G(t)(y - F_t)$, with

$$G(t) := \left(\left(\frac{\partial F_t}{\partial \theta(t)}\right)_p^T \left(\frac{\partial F_t}{\partial \theta(t)}\right)_q\right)_{p,q=1}^{nM} \quad (5)$$

where $\frac{\partial F_t}{\partial \theta(t)} = \left(\frac{\partial (f(X_1, \theta(t)))_1}{\partial \theta(t)}, \ldots, \frac{\partial (f(X_1, \theta(t)))_M}{\partial \theta(t)}, \ldots, \frac{\partial (f(X_n, \theta(t)))_1}{\partial \theta(t)}, \ldots, \frac{\partial (f(X_n, \theta(t)))_M}{\partial \theta(t)}\right)$ is a $P \times nM$ matrix and $P \in \mathbb{N}$ is the number of trainable parameters. $G(t)$ is the gram matrix called the NTK at time $t$ of training.

Now, $G(t)$ is a nonegative definite matrix, since it is a gram matrix. We further claim (from lemma 3.2) that it is a positive definite matrix almost surely (with respect to initialization and input data).

Differentiating the loss, we get

$$\frac{\mathrm{d}}{\mathrm{d}t}\left(||y - F_t||^2\right) = -2(y - F_t)^T G(t)(y - F_t) \leq -2\lambda_0 ||y - F_t||^2 \quad (6)$$

where $\lambda_0 > 0$ is the least eigenvalue of $G(t)$ for $t \in [0, T]$. $\lambda_0 > 0$ is true because of the following argument. The minimum eigenvalue of a matrix varies continuously with the matrix entries. $G_{pq}(t)$ is continuous since $\theta(t)$ varies continuously with $t$ and $\sigma$ is continuous. Thus, by extreme value theorem applied on the compact interval $[0, T]$, the minimum eigenvalue is achieved at some $t_0 \in [0, T]$. Say the minimum eigenvalue of $G(t)$ at $t_0$ is $\lambda_0$. We know $\lambda_0 > 0$ due to lemma 3.2.

Rearranging, we get $\frac{\mathrm{d}}{\mathrm{d}t}\left(e^{2\lambda_0 t}||y - F_t||^2\right) \leq 0$. Thus, putting in $t = 0$ and $t = s \leq T$, we get $||y - F_s|| \leq e^{-\lambda_0 s}||y - F_0||$. $\qquad \square$

**Corollary 3.1.1** (Extension to ReLU activation). *The above linear convergence result holds for ReLU activation as well.*

*Proof.* Let $F_{\alpha,t}, G_\alpha(t)$ be the output vector and gram matrix for a leaky ReLU network with hyperparameter $\alpha \leq 1$ at time $t$. Let $F_t$ be the output vector for the same network but with ReLU activation instead of leaky ReLU, at time $t$. Now, note that

$$(y - F_{\alpha,t})^T G_\alpha(t)(y - F_{\alpha,t}) \geq (y - F_{\alpha,t})^T \tilde{G}(t)(y - F_{\alpha,t})$$

where $\tilde{G}(t)$ stands for the gram matrix of the linearized network. This holds because $G_\alpha(t)_{pq} \geq \tilde{G}(t)_{pq}$ for all $p, q \in [nM]$ since leaky ReLU is bounded below by identity function. Since equation 4 holds for any a.e. differentiable function and leaky ReLU converges to ReLU,

$$\frac{\mathrm{d}}{\mathrm{d}t}\left(||y - F_t||^2\right) = -2(y - F_t)^T G(t)(y - F_t) = -2\limsup_{\alpha \to 0}(y - F_{\alpha,t})^T G_\alpha(t)(y - F_{\alpha,t})$$

$$\leq -2\limsup_{\alpha \to 0}(y - F_{\alpha,t})^T \tilde{G}(t)(y - F_{\alpha,t}) = -2(y - F_t)^T \tilde{G}(t)(y - F_t)$$

Now, following the same arguments as in theorem 3.1, we reach the conclusion for ReLU function as well. $\qquad \square$

**Corollary 3.1.2** (Extension to sigmoid activation). *The above result holds for sigmoid activation as well.*

*Proof.* Applying Jackson's inequality (see theorem B.2) to sigmoid, we get that there exists a sequence of polynomials $(p'_n)$ such that $||\sigma - p'_n||_\infty \leq \frac{1}{4}$, since $|\sigma'| \leq 1/4$, $\pi/2 < 1$ and $n + 1 > 1$. Note that Jackson's inequality holds for differentiable functions over a compact domain, but we can always construct the polynomial such that Jackson's inequality holds in some $[-M, M]$ for $(p'_n)$.

We then define our new sequence of piecewise polynomials $(p_n)$ that converges to $\sigma$ as follows : $p_n = p'_n$ over $[-M, M]$, $p_n(x) = \sigma(M)$ for $x \geq M$ and $p_n(x) = \sigma(-M)$ for $x \leq -M$, where $M > 0$ is chosen such that $|\sigma(x) - 1| < \varepsilon$ for $x \geq M$ and $|\sigma(x)| < \varepsilon$ for $x \leq -M$, for an appropriate $\varepsilon > 0$ so as to ensure uniform convergence. This can be done due to the horizontal asymptotes of sigmoid.

Let $F_{n,t}, G_n(t)$ be the output vector and gram matrix for a network with activation $p_n$ at time $t$. Let $F_t$ be the output vector for the same network but with sigmoid activation instead of $p_n$, at time $t$. Now, note that

$$(y - F_{n,t})^T G_n(t)(y - F_{n,t}) \geq (y - F_{n,t})^T \tilde{G}(t)(y - F_{n,t})$$

where $\tilde{G}(t)$ stands for the gram matrix of any piecewise nonzero polynomial such that it is always at least $1/4$ below $p_n$, since $\sigma > 0$ which implies $p_n < 1/4$. For example the piecewise polynomial $q(x) := x - \frac{1}{4}$ for $x \leq \frac{1}{4}$, and $\frac{1}{4} - x$ for $x \geq \frac{1}{4}$.

Since equation 4 holds for any a.e. differentiable function and $p_n$ converges to $\sigma$,

$$\frac{\mathrm{d}}{\mathrm{d}t}\left(||y - F_t||^2\right) = -2(y - F_t)^T G(t)(y - F_t) = -2\limsup_{n \to \infty}(y - F_{n,t})^T G_n(t)(y - F_{n,t})$$

$$\leq -2\limsup_{n \to \infty}(y - F_{n,t})^T \tilde{G}(t)(y - F_{n,t}) = -2(y - F_t)^T \tilde{G}(t)(y - F_t)$$

Now, following the same arguments as in theorem 3.1, we reach the conclusion for sigmoid function as well. $\qquad \square$

We shall now complete the proof of theorem 3.1 by proving that $G(t)$ remains positive definite throughout training for almost all initializations and input data.

**Lemma 3.2** (Positive definiteness of NTK). $G(t)$ *is a positive definite matrix almost surely (with respect to initialization and input data) for all $t \geq 0$.*

*Proof.* We know that $G(t)$ is positive definite iff $\left\{ \left( \frac{\partial F_t}{\partial \theta(t)} \right)_1, \ldots, \left( \frac{\partial F_t}{\partial \theta(t)} \right)_{nM} \right\} \subset \mathbb{R}^P$ is linearly independent. Note that this set is always linearly dependent if $nM > P$. Thus, we shall assume that $nM \leq P$. We therefore need the function to be overparametrized with respect to the training sample size.

If this set is linearly dependent at $t = 0$, then all sub-square matrices of $\left( \left( \frac{\partial F_0}{\partial \theta(0)} \right)_1 \cdots \left( \frac{\partial F_0}{\partial \theta(0)} \right)_{nM} \right)$ $\in \mathbb{R}^{P \times nM}$ will be singular. Note that the determinants of these matrices are piecewise polynomials in the parameters and the dataset and $\sigma$ is zero only at finitely many points, since it is a piecewise nonzero polynomial. Thus, the determinants are nonzero for Lebesgue-almost all values of the parameters $\theta(0)_1, \ldots, \theta(0)_P$ and the datasets $X_1, \ldots, X_n$ (Caron, 2005). Let $\theta_{\text{degen}}$ denote the set of those parameters for which the determinants are zero, union the set of degenerate parameters from the proof of lemma 2.1. Due to the continuous distribution of initial parameters and input data, we conclude that $\mathbb{P}(\theta(0) \in \theta_{\text{degen}}) = 0$.

We shall now show that $\mathbb{P}(\theta(t) \in \theta_{\text{degen}}) = 0$ for all $t > 0$. From lemma 2.1, we know that the GF solution flow is a piecewise diffeomorphism over the initial conditions space minus a zero measure set. Thus, it maps zero probability sets to zero probability sets (see theorem B.4). Now, we have shown that with respect to initialization and input data, $\mathbb{P}(\theta_{\text{degen}}) = 0$. Thus, from above arguments, $\mathbb{P}(\Phi_{-t}(\theta_{\text{degen}})) = 0$, which implies that $\mathbb{P}(\theta(0) | \Phi_t(\theta(0)) \in \theta_{\text{degen}}) = \mathbb{P}(\theta(t) \in \theta_{\text{degen}}) = 0$. $\qquad \square$

*Remark.* An analogous proof holds for matrix-valued inputs/outputs by flattening the input/output matrix into a vector. We omit the proof for that case for the sake of notational simplicity.

**Corollary 3.2.1** (Linear convergence of specific architectures). *Conclusion of theorem 3.1 holds for the following forms of $f$ (as defined in equation 1):*

1. *(DNN) $f : \mathbb{R}^N \times \mathbb{R}^P \to \mathbb{R}^M$ with $g_i : \mathbb{R}^{N_{i-1}} \times \mathbb{R}^P \to \mathbb{R}^{N_i}$, defined by $g_i(X, \theta) = W_i X + B_i$ for all $X \in \mathbb{R}^{N_{i-1}}$ with $W_i \in \mathbb{R}^{N_i \times N_{i-1}}, B_i \in \mathbb{R}^{N_i}$, for all $i \in [L]$, $N_0 = N, N_L = M$ and $\theta \in \mathbb{R}^P$ denoting the vector of all weights and biases. Over-parametrization requires $P = \sum_{i=1}^{L} N_i(N_{i-1} + 1) \geq nM$.*

2. *(ResNet) $f : \mathbb{R}^N \times \mathbb{R}^P \to \mathbb{R}^N$ with $g_i : \mathbb{R}^N \times \mathbb{R}^P \to \mathbb{R}^N$, defined by $g_i(X, \theta) = X + W_i X + B_i$ for all $X \in \mathbb{R}^N$ with $W_i \in \mathbb{R}^{N \times N}, B_i \in \mathbb{R}^N$, for all $i \in [L]$ and $\theta \in \mathbb{R}^P$ denoting the vector of all weights and biases. Over-parametrization requires $P = L(N^2 + N) \geq nN$.*

3. *(GCN) $f : \mathbb{R}^{m \times N} \times \mathbb{R}^P \to \mathbb{R}^{m \times M}$ with $g_i : \mathbb{R}^{m \times N_{i-1}} \times \mathbb{R}^P \to \mathbb{R}^{m \times N_i}$, defined by $g_i(X, \theta) = \hat{D}^{1/2} \hat{A} \hat{D}^{1/2} X W_i + B_i$ for all $X \in \mathbb{R}^{m \times N_{i-1}}$ with $W_i \in \mathbb{R}^{N_{i-1} \times N_i}, B_i \in \mathbb{R}^{m \times N_i}$, for all $i \in [L]$, $N_0 = N, N_L = M$, $g_L : \mathbb{R}^{m \times N_{L-1}} \to \mathbb{R}^m$ and $\theta \in \mathbb{R}^P$ denoting the vector of all weights and biases. $\hat{A} = A + \mathbb{I}_m, \hat{D} = diag\left( \sum_{j=1}^{m} \hat{A}_{1j}, \ldots, \sum_{j=1}^{m} \hat{A}_{mj} \right)$ where $A$ is the adjacency matrix of the graph over $m$ vertices. Over-parametrization requires $P = \sum_{i=1}^{L} N_i(N_{i-1} + m) \geq nmM$.*

*In each architecture, $\sigma$ can be any piecewise nonzero polynomial activation like leaky ReLU or parametric ReLU, or it can be the commonly used ReLU or sigmoid.*

## 4 EXPERIMENTS

We use synthetic data to support our findings. For all architectures, data is generated uniformly from a ball in $\mathbb{R}^N$ with the $\mathbb{R}^M$-valued labels being generated normally. This is done in concordance to Zhang et al. (2017). We use random synthetic data instead of real-life data to stress-test our theory. Demonstrating that models are capable of fitting to random noise under our over-parametrization scheme provides strong validation of our result.

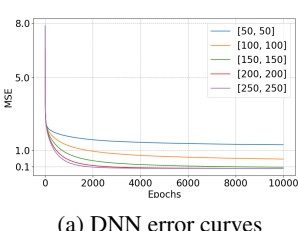 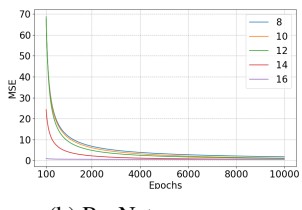 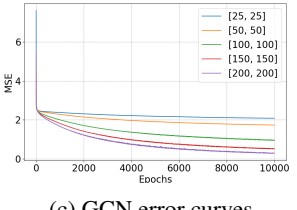

(a) DNN error curves  (b) ResNet error curves  (c) GCN error curves

Figure 3: Results on synthetic data.

Leaky ReLU activation (Maas et al., 2013) is used everywhere, with the initialization being the default Kaiming-He Uniform initialization (He et al., 2015). We perform 'ablation' by varying the number of parameters from being less than $nM$ to slowly being overparametrized. The training curves verify the hypothesis that as we increase overparametrization, these architectures converge closer to linearly in training loss.

The experimental setups are as follows. To simulate GF, a very small learning rate is taken (lr= $10^{-5}$) with a large number of epochs (10,000). Each model is trained over a training sample of size 1000. For DNN, the input dimension is 500 with output dimension 50 and the model is trained for varying hidden layer sizes - starting from two hidden layers of 50 neurons each ([50,50]) and ending with two hidden layers of 250 neurons each ([250,250]). For ResNet, the input and output dimensions are 100 and the model is trained for varying number of hidden layers - starting from 8 and ending with 16. We had to remove showing the first 100 epochs of ResNet because the initial error was very large, causing the graph to become incomprehensible - the exponential decay is nevertheless visible. For GCN, the input dimension is 500 with output dimension 50 and the model is trained for varying hidden layer sizes - starting from two hidden layers of 25 neurons each ([25,25]) and ending with two hidden layers of 200 neurons each ([200,200]). The underlying graph used is a $k$-nearest neighbours graph on the input data with $k = 10$.

## 5 CONCLUSION AND FUTURE DIRECTIONS

This paper studies linear convergence of gradient flow to a global minimum almost surely (with respect to initial weights and input distributions) for any function that can be written as a composition of polynomials and piecewise nonzero polynomials, given that the function is sufficiently overparametrized ($P \geq nM$). In particular, we prove linear convergence of GF to a global minimum almost surely for a large variety of neural networks (like DNNs, GCNs, ResNets, etc.) with activations like leaky ReLU, parametric ReLU, ReLU and sigmoid, given that the function is sufficiently overparametrized ($P \geq nM$).

We prove our main result by exploiting properties of the NTK and how its lowest eigenvalue behaves under the operation of the GF ODE. We extend to ReLU and sigmoid by approximating them uniformly with piecewise nonzero polynomials. We shall now present some future directions.

Future works can try to extend this paper's results more by doing a comparative study of architectures on a case-by-case basis. For example, it has been shown by Shi et al. (2024) that transformers need lesser parameters than DNNs to approximate hierarchial composition models (composed of Korobov/Barron functions), for the same error. Showing that transformers have better convergence properties (in the sense of either weaker assumptions or stronger conclusions) would be an interesting line of work. Moreover, we rely on the loss being the squared loss. This is the most common loss used for regression, but not so much for classification. Future works can try extending this result to other losses like cross-entropy.

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

## A  LLM USAGE

LLMs were used to polish writing at places - like making sentences more concise or formal. Searching for previous works was done by employing both search engines and LLMs. Some standard proofs, results and code for the experiments were also generated by LLMs to save time.

## B  AUXILLARY RESULTS

**Theorem B.1.** *The solution curve of the GF equation 2 reaches a subspace $S$ of codimension $k \geq 2$ for almost no initializations.*

*Proof.* Define the set of bad initializations as

$$A := \{\theta_0 \in \mathbb{R}^P : \exists t \in [0, T] \text{ such that } \Phi_t(\theta_0) \in S\} \tag{7}$$

We shall show that $\lambda^P(A) = 0$. Define the earliest time that the curve hits $S$, for $\theta \in A$ by

$$t_0(\theta) := \inf\{t > 0 : \Phi_t(\theta) \in S\} \tag{8}$$

Define the $r$-neighbourhood around $S$ for any $r > 0$ by

$$N_r(S) := \{\theta \in \mathbb{R}^P : ||x - S|| < r\} \tag{9}$$

Since the flow is Lipschitz continuous in $t$, $||\Phi_t(\theta_0) - \Phi_q(\theta_0)|| \leq L|t - q|$ for all $t, q \in [0, T]$. Choose a finite rational grid $Q_r$ over $[0, T]$ with spacing $\tau \leq r/L$. Note that we cannot choose a finer spacing as that will allow $\Phi_q(\theta) \notin N_r(S)$, which will cause a contradiction later on in the proof.

Thus, number of points in $Q_r$ is $|Q_r| \leq TL/r$. Thus, for all $t \in [0, T]$, we can find $q \in Q_r$ such that $||\Phi_t(\theta_0) - \Phi_q(\theta_0)|| \leq r$.

Furthermore, since $\Phi_q$ is $C^1$ almost everywhere, $\lambda^P(\Phi_q^{-1}(N_r(S))) = \int_{N_r(S)} |\det D\Phi_q^{-1}(y)| dy \leq J\lambda^P(N_r(S))$ for some constant $J > 0$. So, putting all of the above results together,

$$\lambda^P(A) \leq \sum_{q \in Q_r} \lambda^P(\Phi_q^{-1}(N_r(S))) \tag{10}$$

$$\leq |Q_r| J \lambda^P(N_r(S)) \tag{11}$$

$$= \mathcal{O}(r^{k-1}) \tag{12}$$

where last line follows from the fact that volume of $N_r(S)$ is $\mathcal{O}(r^k)$ as $r \to 0$, which follows from Weyl's tube formula (see Gray (2003)). Finally, taking $r \to 0$, we get our conclusion. $\qquad\square$

**Theorem B.2** (Jackson's inequality)**.** *Let $k \in \mathbb{N}$ and $a < b$. Any $k$-times continuously differentiable function $f : [a, b] \to \mathbb{R}$ can be uniformly approximated by a sequence of polynomials $(p_n)$ such that*

$$||f - p_n||_\infty \leq \frac{\pi}{2} \frac{|f^{(k)}|}{(n+1)^k} \tag{13}$$

*Proof.* See Cheney (1998). $\qquad\square$

**Theorem B.3.** *Solution of the GF equation 2 does not blow up in finite time, ie. $||\theta(s)|| < \infty$ for all $s \in (0, \infty)$.*

*Proof.* Note that

$$\frac{\mathrm{d}L(\theta(t))}{\mathrm{d}t} = \nabla L(\theta(t))^T \left( \frac{\mathrm{d}\theta(t)}{\mathrm{d}t} \right) = -||\nabla L(\theta(t))||^2 \tag{14}$$

Thus,

$$\int_0^s ||\nabla L(\theta(t))||^2 dt = -\int_0^s \frac{\mathrm{d}L(\theta(t))}{\mathrm{d}t} dt = L(\theta(0)) - L(\theta(s)) \leq L(\theta(0)) < \infty \tag{15}$$

Now,

$$||\theta(s) - \theta(0)|| = \left\lVert \int_0^s \frac{\mathrm{d}\theta(t)}{\mathrm{d}t} dt \right\rVert = \left\lVert \int_0^s \nabla L(\theta(t)) dt \right\rVert \leq \int_0^s ||\nabla L(\theta(t))|| dt \tag{16}$$

Using Cauchy-Schwartz,

$$\left( \int_0^s 1 \cdot ||\nabla L(\theta(t))|| dt \right)^2 \leq s \int_0^s ||\nabla L(\theta(t))||^2 dt \tag{17}$$

Thus,

$$||\theta(s) - \theta(0)||^2 \leq s \int_0^s ||\nabla L(\theta(t))||^2 dt < \infty \tag{18}$$

Thus, $||\theta(s)|| < \infty$ for all $s > 0$. $\qquad\square$

**Theorem B.4.** *Let $f : \mathbb{R}^P \to \mathbb{R}^P$ be a piecewise differentiable function (with pieces $E_1, \ldots, E_p$ such that $\cup_{i=1}^p E_i = \mathbb{R}^P$). Let $A \subset \mathbb{R}^P$ such that $\lambda^P(A) = 0$. Then, $\lambda^P(f(A)) = 0$.*

*Proof.* We know that $f|_{E_i}$ will map zero measure sets to zero measure sets (see Rudin (2006)). Note that $A = A \cap \mathbb{R}^P = A \cap (\cup_{i=1}^p E_i) = \cup_{i=1}^p (A \cap E_i)$. Thus, $\lambda^P(f(A)) \leq \sum_{i=1}^p \lambda^P(f(A \cap E_i)) = 0$. $\qquad\square$

