# OpenReview forum: "Gradient Flow Convergence Guarantee for General Neural Network Architectures"
_ICLR.cc/2026/Conference — ICLR 2026 Conference Withdrawn Submission_

### Official Review · Reviewer_RJhh · 2025-10-16

**Soundness:** 1
**Presentation:** 2
**Contribution:** 1
**Rating:** 0
**Confidence:** 4

**Summary:**

This work aims to provide a unified analysis for gradient flow of neural network training. The major goal is to relax the assumptions on parameter size, and model architectures used in known analysis.

**Strengths:**

The topic--to simplify and unify existing gradient flow analysis--is a valid research topic.

**Weaknesses:**

My major concern is that this work doesn't prove what it claims to prove. Most tools and arguments are standard in GF analysis of neural nets. The key difference is in line 324-329.

This argument basically explains why the authors can greatly simplify the assumptions by previous works, because it basically just assume the spectral of the grammian always has a positive margin. This assumption 1. makes the analysis trivial, 2. was not explicitly stated in theorems.

If one does not make the assumption, then the argument in 324-249 itself is incorrect. In asymptotic regime, one needs to first prove that the convergence time is finite, before proving exponential convergence.

**Questions:**

1. In eq (1), how does g_i, i >1,  depend on theta? This may become clear later in examples, but this equation seems to be incorrect.
2. In figure 2, what does every piece represent?

---

### Official Review · Reviewer_GKYA · 2025-10-16

**Soundness:** 1
**Presentation:** 3
**Contribution:** 1
**Rating:** 0
**Confidence:** 5

**Summary:**

The paper studies the problem of convergence of gradient flow (GF), which is the small stepsize limit of gradient descent, for general neural network architectures (in fact no argument is specific to neural networks, and the approach applies to any smooth optimization problem). The paper aims at showing exponential convergence of GF for overparameterized problems, i.e., where the number of parameters is no less than the number of data.

**Strengths:**

The paper is well-written and cites a number of references in the literature. I truly appreciate the efforts of authors to abstract proofs in order to extract the core arguments. This is an important contribution to the research community, which is sometimes undervalued.

**Weaknesses:**

The proof of the result is incorrect. The most fundamental issue is that the constant $\lambda_0$ in the exponential decay in the main result depends on the time horizon $T$. This is a critical problem, given that any strictly decreasing function $f(t)$ can be written $e^{-\lambda_0(t) t}$ for some $\lambda_0(t) >0$. This fundamentally undermines the argument of the authors.

This issue is usually solved in the literature by lower-bounding the smallest eigenvalue of the NTK on a ball around initialization, then showing that the trajectory cannot leave this ball. Obtaining this lower-bound is architecture-dependent (and also relies on a so-called lazy initialization), and is the lengthiest part of this kind of proofs. To my knowledge, this step cannot be avoided (although I would be happy to see a simple proof for this step that abstracts away from details of the architecture).

A smaller but still substantial issue is that the non-differentiability of the activation function is not handled appropriately. Even though $\sigma$ is differentiable almost everywhere, its composition with itself might not be (see e.g. Lee et al., 2020), while this property is used by the authors (see e.g. line 226).

Finally, previous works already proved linear convergence with overparameterization that is linear in the sample size, see e.g. Marion et al., 2024, for a proof (which follows the approach outlined above for a class of residual neural networks).

Lee et al. On Correctness of Automatic Differentiation for Non-Differentiable Functions, NeurIPS 2020.

Marion et al. Implicit regularization of deep residual networks towards neural ODEs, ICLR 2024.

**Questions:**

See weaknesses.

---

### Official Review · Reviewer_LT1e · 2025-10-27

**Soundness:** 1
**Presentation:** 2
**Contribution:** 1
**Rating:** 0
**Confidence:** 4

**Summary:**

This paper studies the convergence guarantee of gradient flow when applied to train neural network with general architecture. In particular, the paper considers neural networks that can be written as compositions layer-wise functions and a piecewise polynomial activation, and shows that training the neural network over the MSE loss using gradient flow enjoys a linear convergence with rate depending on the minimum eigenvalue of the neural tangent kernel (NTK). The paper provided concrete examples of the activation functions (ReLU and sigmoid) that falls into the scope of the analysis, and backed up the theory with experimental result that shows the convergence of gradient flow.

**Strengths:**

1. The paper aims to tackle an important problem, i.e. the convergence of gradient flow on neural networks. Prior work either require a significant overparameterization, or is restricted to two-layers. The central question in the paper is significant due to its generality and relaxed assumptions.
2. The paper provided results that shows that gradient flow on neural networks with piecewise polynomial activation generates a diffeomorphism over the parameter space.

**Weaknesses:**

1. The main issue with the paper is its correctness. In particular, in the proof of Lemma 2.1 the paper claims that "Since GF cannot reach such a critical point, we need not worry about this case for the inverse problem". Showing that optimization problems will not be trapped at critical points has been a longstanding difficulty, and in this particular case simply making the claim without a formal proof does not constitutes a valid theoretical result.
2. is positive definite, which is a standard result in NTK literatures. Based on this argument, the paper claims that "(the gradient flow) maps zero probability sets to zero probability sets", thus concluding that the NTK during training with have a zero probability of not being positive definite. There are two issues with this argument. First, the claim that "(the gradient flow) maps zero probability sets to zero probability sets" only shows that $\mathbb{P}(\theta\in \theta_{\text{degen}})$ when $\theta$ follows the same probability distribution as the initialization, which is not the case when $\theta(t)$ is generated by the gradient flow function. Second, the paper only tries to show that with zero probability the eigenvalues of $G(t)$ contains a zero. It does not show that all eigenvalues of $G(t)$ are positive.
3. The experiment does not given a strong validation for the theory. As in the analysis, the loss convergence should follow a simple rule of $e^{-\lambda_0t}$ where $\lambda_0$ is no smaller than the minimum eigenvalue of the NTK at initialization. The paper should compare the loss convergence curve with the theoretically predicted curve to better support the theoretical analysis.

**Questions:**

None.

---

### Official Review · Reviewer_BG67 · 2025-11-02

**Soundness:** 1
**Presentation:** 2
**Contribution:** 1
**Rating:** 2
**Confidence:** 4

**Summary:**

The paper claims to give training error bounds under gradient flow for general deep learning architectures.

**Strengths:**

The paper points to classic results in mathematics that might be of use in giving general and elegant proofs for convergence of optimisation methods in deep learning.

**Weaknesses:**

The paper's claims are very strong, but the results shown can at best be viewed as uninspiring. A strict reading of the results shows several outright errors. The errors are because of the hand-wavy nature of usage of several terms and several leaps of logic.

This leads to several isses. Details below. I assume output dimesion $M=1$ for simplicity.

1. Theorem 3.1 (excluding the $\lambda_0 > 0$ ) is already well known. e(t) = NTK(t) @ e(t-1) is a well known result from as early as 2017 (Jacot et al.). If the lowest eigen value of NTK(t) can be bounded above zero, a linear convergence is trivial.

2. The lower bound on the smallest eigen value of NTK(t) is the main technical challenge. This is attempted in Lemma 3.2. This does not seem well reasoned however. e.g. Determinant being a piecewise polynomial of the parameters is correct, but why is it non-zero?  $\sigma(t)$ not being zero, does not mean the same as the partial derivative terms in the matrix not being zero. The simplest counter example is that you get derivative terms $\sigma'(t)$ terms in the elements of the Jacobian. So even if $\sigma(t)$ is bounded away from zero, $\sigma'(t) $ could be zero with non-zero probability over init and data. There could be other issues as well. One immediate fix might be to bound the derivative $\sigma'$ also away from zero. But I am not sure even this fix will work.

3. In Theorem 3.1, what prevents $\lambda_t$ (the minimum eigen value of G(t)) being of the order of 1/t ? In this case, there is no exponential decay  -- $\min_t t \lambda_t $  is a constant. Wiggling around this by saying the upper limit $T$ is fixed, results in a vacuous theorem -- ANY decreasing function $h(t)$ with $h(0)=1$ can be upper bounded  in the domain $[0,T]$ by a decaying exponential $\exp(-\lambda t)$ for some $\lambda > 0$ if $\lambda$ is allowed to depend on $T$. When using constants like $\lambda_0$ in Theorems, either give an explicit formula for it, or make precise what all terms is this constant INDEPENDENT of. Otherwise, most theorems would be vacuous. For this to be a non-vacuous theorem, it should give a lower bound for $\lambda_0$ that is independent of $T, n, M, N, P$ or give an explicit dependence of $\lambda_0$ on these quantities.

4. Taking Corollary 3.1.1. applied to Theorem 1 means that with probability $1$ training a 2 layer ReLU net, with one hidden neuron in the hidden layer with $N=10, n=10, M=1$  gives exponentially decaying error. This is clearly not true -- all data points that are in the "inactive region" of the initial hidden neuron will NEVER have their error reduced. On average half the data points will be in the inactive region. This example follows all the constraints put in the paper on init and data (absolutely continuous).

5. Issues with Corollary 3.1.1 proof: What exactly is $\tilde G$ ? Is it simply the pairwise data inner product matrix? That does not depend on training time $t$ however. Anyways, I don't see how both the statements hold: i.e why is $G_\alpha(t) > \tilde G(t) $? ReLU being bounded below by identity does not directly imply this. And how does that implication give the inequality on the Mahalanobis distances? That would require $G_\alpha - \tilde G$ be postive semi-definite -- this was not proved. A more fundamental reason this is wrong is that, the Theorem statement is just not true for ReLU (see counter example in point 3 above).

6. Potentially, you could have a setting where all of these issues vanish. e.g Gaussian data, LeakyReLU, overparameterisation. In this case, I do believe a linear convergence of gradient flow would be reasonable. In this case, however, the analysis is mostly well-known and follows from pre-existing literature. (e.g Arora et al. 2019 On exact computation with infintely wide network, and any classic literature on least squares minimization). No new insight is present. Under this distribution setting and overparameterization, any reasonable algorithm would optimize efficiently, even just linear models with gradient flow. No new insight into architecture, structure or feature learning is available due to this analysis.

7. There also seems to be several other misunderstandings. Resnet is NOT just adding an identity matrix to the initialisation weight matrix. (In Corollary 3.2.1 DNNs become Resnets by replacing W_i with W_i + I ).

8. Minor issue: Equation 5 G(t) is not a concatenation, it is a summation.

**Questions:**

See weaknesses.

---

### Note · Authors · 2025-11-20

**Comment:**

I would like to thank the reviewers for pointing out the fatal flaw of the theorem being valid in finite time, thus making it vacuous. I shall withdraw this submission and work on the flaw before resubmitting.

**Withdrawal Confirmation:**

I have read and agree with the venue's withdrawal policy on behalf of myself and my co-authors.